# Influence of Foliar Kaolin Application and Irrigation on Photosynthetic Activity of Grape Berries

**Andreia Garrido [1]**, **João Serôdio [2]**, **Ric De Vos [3]**, **Artur Conde [1,4]** and **Ana Cunha [1,4,5,*]**

[1] Centre for the Research and Technology of Agro-Environmental and Biological Sciences (CITAB), University of Minho, Campus de Gualtar, 4710-057 Braga, Portugal; andreiagarrido@sapo.pt (A.G.); arturconde@bio.uminho.pt (A.C.)

[2] Centre for Environmental and Marine Studies (CESAM), University of Aveiro, Campus de Santiago, 3810-193 Aveiro, Portugal; jserodio@ua.pt

[3] Wageningen Plant Research, Wageningen University and Research Centre (Wageningen-UR), PO Box 16, 6700 Wageningen, The Netherlands; ric.devos@wur.nl

[4] Centre of Molecular and Environmental Biology (CBMA), University of Minho, Campus de Gualtar, 4710-057 Braga, Portugal

[5] Centre of Biological Engineering (CEB), University of Minho, Campus de Gualtar, 4710-057 Braga, Portugal

* Correspondence: accunha@bio.uminho.pt; Tel.: +351- 253-604-046

**Abstract:** Climate changes may cause severe impacts both on grapevine and berry development. Foliar application of kaolin has been suggested as a mitigation strategy to cope with stress caused by excessive heat/radiation absorbed by leaves and grape berry clusters. However, its effect on the light micro-environment inside the canopy and clusters, as well as on the acclimation status and physiological responses of the grape berries, is unclear. The main objective of this work was to evaluate the effect of foliar kaolin application on the photosynthetic activity of the exocarp and seeds, which are the main photosynthetically active berry tissues. For this purpose, berries from high light (HL) and low light (LL) microclimates in the canopy, from kaolin-treated and non-treated, irrigated and non-irrigated plants, were collected at three developmental stages. Photochemical and non-photochemical efficiencies of both tissues were obtained by a pulse amplitude modulated chlorophyll fluorescence imaging analysis. The maximum quantum efficiency ($F_v/F_m$) data for green HL-grown berries suggest that kaolin application can protect the berry exocarp from light stress. At the mature stage, exocarps of LL grapes from irrigated plants treated with kaolin presented higher $F_v/F_m$ and relative electron transport rates ($rETR_{200}$) than those without kaolin. However, for the seeds, a negative interaction between kaolin and irrigation were observed especially in HL grapes. These results highlight the impact of foliar kaolin application on the photosynthetic performance of grape berries growing under different light microclimates and irrigation regimes, throughout the season. This provides insights for a more case-oriented application of this mitigation strategy on grapevines.

**Keywords:** light micro-climates; mitigation strategies; kaolin; irrigation; *Vitis vinifera* L.; grape berry tissues; pulse amplitude modulated (PAM) fluorometry; photosynthesis; photosynthetic pigments

## 1. Introduction

Viticulture is a historically important agronomic and socio-economic sector in Portugal. According to the last report from the International Organization of Vine and Wine (OIV), Portugal is the 11th world and 5th European wine producer [1]. With 14 winemaking regions distributed throughout the country, the Vinhos Verdes or Minho region, as well as the Douro and Alentejo, are the major contributors for national exports and growth of this sector [2].

Grapevine is influenced by a complex and interacting system commonly called *terroir*, which, according to the OIV [3], includes specific soil, topography, climate, landscape characteristics and biodiversity features, and interaction with applied vitivini-cultural practices. This complex system influences the canopy microclimate and grapevine physiology and development and, consequently, grape berry quality and the organoleptic properties of its wine, which is typical of each region.

Currently, climate change projections point to a particularly pronounced temperature variation, with an overall increase of up to 3.7 °C by the end of this century, compared to the 1985–2005 reference period [4]. These temperature changes will have great impacts in the Mediterranean wine regions [5]. According to recent investigations using very high resolution bioclimatic zoning, both temperature and dryness are predicted to increase in several economically important Portuguese viticulture regions, including the Vinhos Verdes region [6]. Therefore, Portuguese vineyards will be subject to increased stress due to the interaction of the existing high radiation levels with the foreseen elevated air temperature and drought, which, all together, can have high impact on grapevine phenology, physiology, and productivity. Several of these climate impacts have already been reported, such as: earlier phenological timings and shortenings of the grapevine growing season [7], sunburns in leaves and grape berries [8], reduction of stomatal conductance and decrease of photosynthetic rates, either by stomatal and non-stomatal limitations [9], appearance and/or intensification of grapevine-related pests and diseases [10,11], increased grape sugar concentrations that lead to higher wine alcohol levels, lower acidities, and modification of varietal aroma compounds [12], and higher inter-annual yield and wine production variability [13].

In order to mitigate these adverse climate effects, new short-term measures have recently been implemented in Portuguese vineyards, such as smart irrigation [14,15] and foliar application of kaolin [16]. Vineyards are not traditionally irrigated and there are even restrictions on this practice in some regions, such as the Douro region [15]. However, according to a recent projection model, a 10% reduction in grapevine yield is expected in the Minho region if irrigation is not applied [14]. Kaolin is a white, chemically inert, and non-toxic clay material ($Al_2Si_2O_5(OH)_4$) that can reflect radiation, including photosynthetically active (PAR), ultraviolet (UV), and infrared radiation (IR) [16]. Foliar application of this mineral has become a cost-efficient mitigation strategy to cope with water stress and excessive heat/radiation absorbed by leaves and grape berry clusters, which also proves effective in alleviating negative impacts on grapevines [17–21]. However, the amount and spectral quality of light intercepted by leaves and transmitted/ reflected into the canopy, crucial factors for leaf, and fruit physiology and development [22] are also important aspects to consider when mitigation practices are used.

Previous work done by our group, using pulse amplitude modulated (PAM) chlorophyll fluorescence imaging, has mapped grape berry photosynthesis at a histological level, and revealed both the exocarp and the seed outer integument as the main photosynthetically competent tissues [23]. More recently, we have studied the photosynthetic performance of grape berry tissues from clusters growing at three distinct light microclimates in the canopy and observed microclimate-related differences in their photosynthetic capacity and acclimation status [24]. This led to the hypothesis that, if a specific viticulture practice changes the light reaching the clusters, and alters its light microclimate, it may impact the photosynthetic activity of berry tissues and associated tissue-specific biochemical processes. In fact, foliar kaolin application may have direct implications on light distribution at the whole canopy level, and irrigation is an indirect one, through increased vegetative growth. For instance, it has already been shown that kaolin application generally reduces the photosynthetic rates of individual leaves in other agricultural crops (e.g., apple, almond, and walnut canopies) [25,26], due to a 20%–40% increase in the reflection of PAR [27]. However, the photosynthesis of the whole canopy remained unaffected or even increased (9%), because of the better light distribution within the canopy [28–30]. In another study, decreased photosynthesis was observed in the inner leaves of irrigated grapevines due to higher vegetative growth [31]. While the function of photosynthesis in fruits is still poorly

understood, it can be linked with primary and secondary metabolomic pathways [32,33]. Therefore, any effect on photosynthesis may impact grape berry development and composition.

Therefore, the main objective of the present work was to evaluate the effects of foliar kaolin application on the photosynthetic activity of grape berry tissues from clusters growing at two distinct microclimates, which include high light (HL) and low light (LL) microclimates, of irrigated and non-irrigated grapevines, during the season.

## 2. Materials and Methods

### 2.1. Site Description, Applied Treatments, and Sampling

Grape berry samples were collected in 2018 from field-grown 'Alvarinho' cultivar grapevines (*Vitis vinifera* L.) in the commercial vineyard *Quinta Cova da Raposa* in the Demarcated Region of Vinho Verde, Braga, Portugal (41°34′16.4″ N, 8°23′42.0″ W). The vineyard is managed by following standard cultural practices applied in organic farming, and is arranged in terraces along a granitic hillside with high drainage. The vine training system applied for this cultivar follows the settings of Sylvoz (Simple Ascending and Recumbent Cord). The sector selected for the trial was located on a hill with NW-SE orientation and the vineyard rows with a NE-SW orientation. The treatments applied were: kaolin (K) and non-kaolin (NK) application on leaves, and irrigation (I) and non-irrigation (NI), in a complete factorial design (four treatment combinations) with two blocks, each with three to four vines per combination treatment (Figure 1B). A suspension of 5% (w/v in water) kaolin (EPAGRO®, Sunprotect, Alverca do Ribatejo, Portugal) was applied twice on leaves on both sides of the rows. On July 6 and 27, corresponding to four weeks after anthesis (WAA) or BBCH-73 (BBCH-scale used for grapes by Lorenz et al. [34]) and seven WAA or BBCH-77, respectively. Irrigation of half of the plants, started on July 26 (seven WAA, BBCH-77), (Figure 1A,B). Water was applied by drip irrigation with one dripper per vine and a drip line placed approximately 80 cm above the soil. Irrigation occurred every three days, once a day either early in the morning or late in the afternoon, for 2 h, with an average dripper capacity of $5.5 \pm 1.6$ L h$^{-1}$ ($n = 12$ randomly selected drippers, $\pm$ SD). Clusters with contrasting light exposure were also selected to harvest grape berries during their development. These were called low light (LL) and high light (HL) clusters. LL clusters grew in the shaded inner zones of the canopy, which were exposed only to diffuse, reflected, and transmitted light, while HL clusters were exposed to direct or reflected sunlight most of the day. Six independent sub-clusters (three per block), each containing 15–20 grape berries, were collected randomly from clusters growing at each of the experimental conditions (four treatments × 2 microclimates) from the southeast side of rows. Berries were harvested in the morning (9–10 a.m.) at three distinct developmental stages: Green (16 July, 6 WAA, BBCH-75), *Véraison* (29 August, 12 WAA, BBCH-83), and Mature (17 September, 15 WAA, BBCH-89). The material was transported in refrigerated boxes to the Center for Environmental and Marine Studies (CESAM) laboratory and used within 2–6 h for imaging fluorometry experiments. For other assays, berries were immediately frozen in liquid nitrogen and stored at −80 °C.

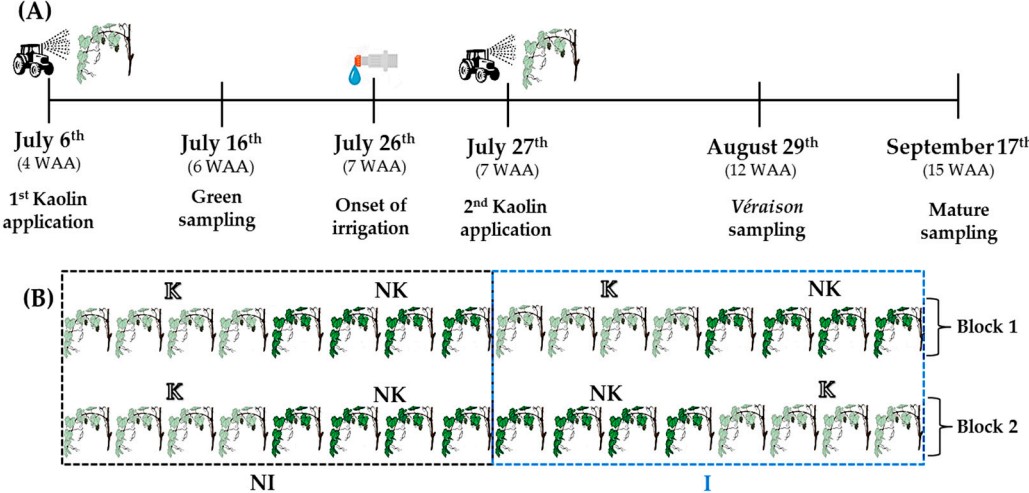

**Figure 1.** (**A**) Timeline of the grape growing season depicting the sampling times, foliar kaolin application dates, and the onset of irrigation. (**B**) Scheme of treatment combinations applied in the field: irrigation (I)/ non-irrigation (NI) × kaolin (K)/ non-kaolin (NK). (WAA - weeks after anthesis).

### 2.2. Light Intensity and Temperature Measurements for Microclimate Characterization

In order to characterize the microclimates in the vicinity of the growing clusters of all experimental conditions, light intensities and temperatures were registered on cloudless days (mean of 1500 ± 300 µmol photons m$^{-2}$ s$^{-1}$), between 15 h and 17 h, at green and mature stages, as described by Garrido et al. [24]. The light intensity (µmol photons m$^{-2}$ s$^{-1}$) was measured with a radiometer (LI-COR, LI-250 Light Meter, Lincoln, NE, USA) and the temperature (°C) was measured with an infrared digital thermometer (Infrared, DT8380, Beijing, China). Both parameters were determined in the frontal region of the clusters (LL and HL), at the southeast side of the row, and in full sun-exposed leaves. The devices were placed perpendicularly to the plant organ (cluster or leaf). The light sensor was placed on the organ surface facing outward, which registered the light intensity reaching at this point, and the thermometer was pointing to the organ at a distance of about 15 cm, which registered an average organ temperature. Sixteen replicate measurements per treatment were considered on randomly selected vines.

### 2.3. Kaolin Film Transmittance and Reflectance

Transmittance and reflectance spectra were obtained using a spectrometer (USB2000-VIS-NIR, grating #3, Ocean Optics, Duiven, The Netherlands), connected to a 400 mm-diameter fiber optic (QP400-2-VIS/NIR-BX; Ocean Optics), and recorded using the spectral acquisition software Spectra Suite (Ocean Optics, https://oceanoptics.com/). The transmittance spectrum was obtained by spreading a 5% (w/v) kaolin suspension prepared in 70% ethanol (allowing fast evaporation to prevent solvent interference), over a glass plate, which simulated particle distributions similar to those observed in the field. A halogen lamp was placed underneath to illuminate the spectrometer sensor positioned 3 cm above the glass plate. Different spectra were obtained from different areas randomly ($n = 3$). The reflectance spectrum was obtained, according to Dinis et al. [18], by pointing the fiber optics perpendicularly to the surface of collected leaves illuminated by the same halogen lamp. Three independent spectra were obtained from different leaf regions of both kaolin-treated and non-treated leaves. Transmittance and reflectance spectra were recorded for the 370–900 nm spectral range, with a spectral resolution of 0.33 nm. The transmittance spectra were expressed as a percentage of the controls (glass). Reflectance spectra were normalized to the spectrum reflected from a reference white panel (WS-1-SL Spectralon Reference Standard, Ocean Optics).

## 2.4. Chlorophyll Fluorescence Analysis

The photosynthetic activity of grape berry tissues was assessed as described by Garrido et al. [24]. For this, an imaging chlorophyll fluorometer was used (*Open FluorCAM 800 MF*; Photon Systems Instruments, Drásov, Czech Republic), which was comprised of four $13 \times 13$ cm LED panels emitting red light (emission peak at 621 nm, 40-nm bandwidth) and a 2/3 inch CCD camera (CCD381, Beijing, China) with a F1.2 (2.8–6 mm) objective. Two of the LED panels provided modulated measuring light ($<0.1$ µmol m$^{-2}$ s$^{-1}$) and the other two provided saturating pulses ($>7500$ µmol m$^{-2}$ s$^{-1}$, 0.8 s). Chlorophyll fluorescence images were captured and processed using FluorCam7 software (Photon Systems Instruments, Drásov, Czech Republic).

In a dark cabinet, exocarps and seeds were separated from dark-adapted berries and disposed in $8 \times 8$-well plates filled with water. Two independent plates were prepared for each microclimate (LL and HL), with each comprising all treatments and tissues. Exocarps and seeds were placed alternately in three rows each, using two columns per treatment, in a total of 12 biological replicates per condition and tissue ($n = 3 \times 2 \times 2 = 12$). Each plate was subjected to the experiments described below.

The maximum quantum efficiency of photosystem II [$F_v/F_m = (F_m - F_0)/F_m$], which is a chlorophyll fluorescence parameter that reflects the probability of electrons being transferred from the PSII reaction center for the transport chain of electrons by quanta absorbed [35,36], was computed following a saturation pulse (SP). The isolated tissues were then acclimated to an actinic light (AL) of 200 µmol photons m$^{-2}$ s$^{-1}$ for 15 min, and, after a new SP, the effective quantum yield of PSII [$\Phi_{II} = (F'_m - F_s)/F'_m$] was computed. This parameter correlated with the quantum yield of $CO_2$ fixation in a wide range of physiological conditions [37]. From $\Phi_{II}$ and PFR (photosynthetic photon fluence rates) (200 µmol photons m$^{-2}$ s$^{-1}$), the relative electron transport rate through PSII ($rETR_{200} = \Phi_{II} \times PFR$) was calculated. Then, tissues were exposed to 1500 µmol photons m$^{-2}$ s$^{-1}$ for 15 min, with an SP being applied every 3 min. The last $F'_m$ values (at 15 min) were used to calculate the non-photochemical quenching [$NPQ = (F_m - F'_m)/F'_m$].

## 2.5. Analysis of Chlorophylls and Carotenoids by High Performance Liquid Chromatography Coupled to A Photodiode Array Detector (HPLC-PDA)

The extraction procedure was adapted from Fraser et al. [38]. Freeze-dried material (20 mg) of grape berry tissues, which includes exocarp and seed, was extracted in 1.8 mL of chloroform/methanol (1:1) (chloroform - Emsure®, Darmstadt, Germany, methanol - Biosolve®, Dieuze, France) with both 0.1% (w/v) butylated hydroxytoluene (BHT, Sigma®, Zwijndrecht, The Netherlands) as an antioxidant and Sudan 1 (0.5 µg mL$^{-1}$) as the internal standard (IS). The samples were vortexed (10 s), kept on ice for 30 min (vortexed in between), and then sonicated for 15 min (Branson®, 3510 Ultrasonic Cleaner, Danbury, CT, USA). These steps were performed twice. After that, the samples were centrifuged at $16{,}100\times g$ (Eppendorf®, Centrifuge 5415 R, Hamburg, Germany) and the supernatant (approx. 1200 µL) was transferred to a new Eppendorf tube with a perforated lid. The samples were dried for 1 h in a Speedvac (Savant®, SC100, Schiphol, The Netherlands) and then stored at $-80$ °C until the next steps. Prior to high performance liquid chromatography (HPLC) analysis, samples were dissolved in 200 µL ethylacetate solution containing 0.1% (w/v) BHT, sonicated (10 min), and then centrifuged as above. Samples were protected from light and kept on ice during all of these procedures. The supernatant (180 µL) was transferred to amber-colored 2 mL HPLC vials with a glass insert and sealed.

The HPLC-PDA procedure was adapted from Mokochinski et al. [39]. The samples (20 µL) were analyzed using an HPLC (Waters Alliance e2695 Separations Module, Milford, MA, USA) coupled to a photodiode array detector (PDA) (Waters 2996) over the 240 to 700 nm UV/Vis range. Separation was performed on a reverse-phase $C_{30}$ column ($250 \times 4.6$ mm i.d., S-5 µm - YMC Carotenoid, Komatsu, Japan) kept at 35 °C with a flow rate of 1.0 mL min$^{-1}$. The compounds were identified based on comparisons of retention times and absorption spectra (240 to 700 nm) with authentic standards.

*2.6. Statistical Analysis*

Results were statistically analyzed using Analysis of Variance tests (two-way ANOVA), followed by post hoc multiple comparisons using the Bonferroni test whenever the factors had significant effects (GraphPad Prism version 5.00 for Windows, GraphPad Software, La Jolla, CA, USA). Significant differences ($p \leq 0.05$) between sample groups are indicated with different letters. Notation with an asterisk means that only one factor (kaolin or irrigation) was significant.

## 3. Results and Discussion

*3.1. Climatic Conditions and Microclimate Characteristics*

In order to characterize the climate during the growing season at the study site (Braga), we used the official information available from the *Instituto Português do Mar e da Atmosfera* (IPMA) [40], to determine the temperatures and total precipitation during 2018 (Figure S1). This growing season was atypical from a climatic point of view, with a relatively cold and extremely dry winter, which caused a delay of sprouting/flowering for two to three weeks [41], and a relatively cold spring with rainy periods during the vegetative growth of the grapevines.

To characterize the microclimates for the LL and HL berry clusters (two *a priori* selected distinct light microclimates within the canopy), measurements of light intensities and temperatures were performed at the cluster level, at two time points during the growing season, i.e., when the berries were still green (green stage) and, two months later, when the berries were at their mature stage of ripening (Figures 2 and 3). Figure 2 depicts the average light intensities at LL and HL clusters growing under the different experimental conditions: i.e., irrigation/non irrigation (Figure 2a,c) and with kaolin/without kaolin (Figure 2b,d). The two microclimates were clearly distinct at both ripening stages, with HL clusters receiving about three-fold more light than LL clusters. At the green stage, i.e., before the onset of irrigation (Figure 1A), no significant differences were detected between the two sets of plots assigned to the subsequent irrigation experiment (four plots for irrigated (I) plants, i.e., 2 × NK-I and 2 × K-I) and four plots for non-irrigated (NI) control plants (2 × NK-NI and 2 × K-NI), see Figure 1B) (Figure 2a), which reveals that there were no plot-related effects on microclimate light intensity. At this early ripening stage, and with the adopted measurement procedure, no differences were detected with respect to light intensities reaching the berry clusters due to foliar kaolin application (Figure 2b). However, at a mature stage, both irrigation and kaolin had a small but significant effect on the light intensity reaching the HL clusters (Figure 2c,d). Irrigation slightly reduced the light intensity, likely due to the better vegetative growth of the plants, while foliar kaolin application increased it, likely due to an increased light reflection to both the interior and lower levels of the canopy. In the LL clusters, these effects of irrigation and kaolin on light intensity were not observed, at this time of day.

HL grapes consistently experienced higher temperatures than LL ones (Figure 3), and both I and K treatments exerted significant and contrasting effects on this microclimate parameter, mainly at the mature stage. Again, and consistent with what was observed and discussed above for light intensity, no effect was detected for I treatment on the grape berry temperature at the green stage (before the onset of irrigation) (Figure 3a).

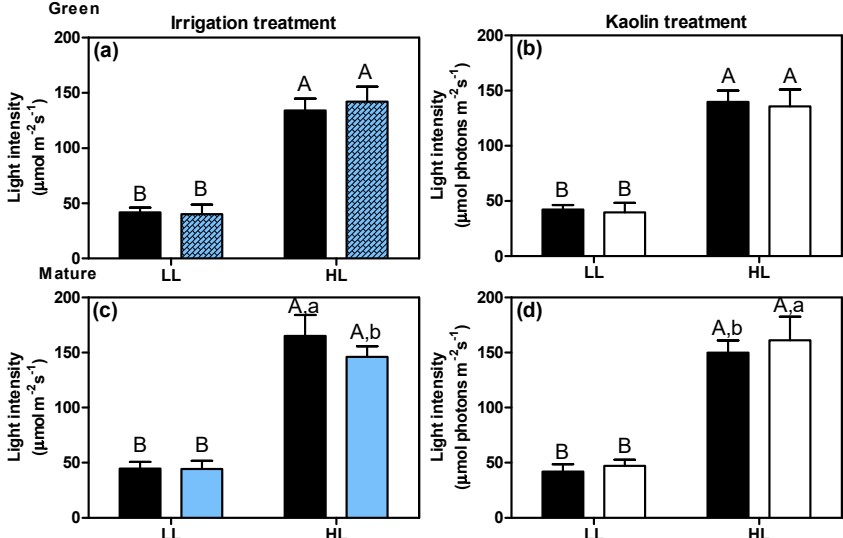

**Figure 2.** Light intensities received by LL and HL clusters at the green stage (**a**,**b**) and the mature stage (**c**,**d**), for plants with irrigation (blue columns, note: the textured blue columns at green stage i.e., before the onset of irrigation, represent the measurements in the plots that were later irrigated) and foliar kaolin application (white columns). Black columns correspond to the respective controls. Values represent means with a standard deviation (*n* = 16 plants). Statistical notation: per ripening stage, different capital letters refer to significant differences (two-way ANOVA, *p* ≤ 0.05) between the two light microclimates within the same plant treatment, and different lowercase letters for differences between treatments within each light microclimate. If the respective factor did not have a significant effect, the lowercase letters were omitted.

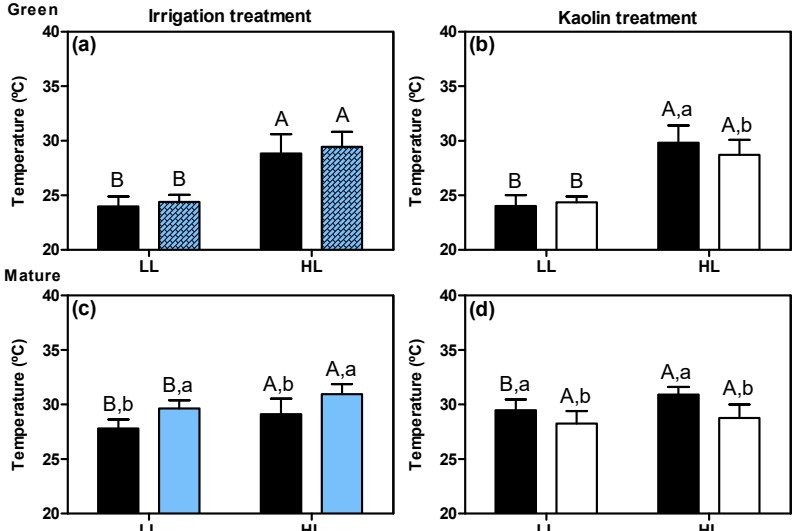

**Figure 3.** Temperatures of LL and HL clusters at the green stage (**a**,**b**) and the mature stage (**c**,**d**), for plants with irrigation (blue columns, note: the textured blue columns at the green stage i.e., before the onset of irrigation, represent the measurements in the plots that were later irrigated) and foliar kaolin application (white columns). Black columns correspond to the respective controls. Values represent means with standard deviation (*n* = 16 plants). Statistical notation: per ripening stage, different capital letters refer to significant differences (two-way ANOVA, *p* ≤ 0.05) between the two light microclimates within the same plant treatment, and different lowercase letters for differences between treatments within each light microclimate. If the respective factor did not have a significant effect, the lowercase letters were omitted.

The Kaolin application led to a significant decrease in the temperature of the HL clusters at both green and mature stages (e.g., at this latter stage—from 31 °C to 28.7 °C), and of the LL clusters at the mature stage only (e.g., from 29.5 °C to 28 °C) (Figure 3b,d). The fact that LL clusters' temperature at the green stage were not affected by kaolin was likely related to the relatively low air temperature at this time of the growing season (early July, Figure S1). Thus, kaolin application on the leaves may increase the incident light by increasing the light reflection inside the canopy (Figure 2d), while maintaining a cooler microclimate for the growing berries, especially during the hot summer days, independently of the irrigation regime. This demonstrates one of the advantages of this mitigation strategy at the grape berry level. The kaolin solution sprayed on the leaves also resulted in leaf temperature reduction (Figure S2d). The present results are in agreement with previous studies in both grapevine leaves and berries [42,43], as well as other crops [26,27]. Therefore, it is likely that kaolin applied to leaves provides cooler temperatures throughout the grapevine by reducing the total amount of radiation transmitted into the canopy. This is also shown by thermal imaging in apple trees [30]. Furthermore, different training systems of the vineyard might influence the light intensities and temperatures inside the canopy [44,45]. For instance, the vine canopy was denser in our previous study in 2015 [24], which resulted in an LL microclimate characterized by much lower light reaching the clusters compared to that in the present study, with major impacts on grape berry photosynthetic competence.

At the mature stage, plant irrigation resulted in a significant increase in the grape temperature of both LL and HL clusters (Figure 3c). This increase was unexpected, since a previous study reported lower berry temperature as a result of irrigation, rather than a higher temperature [46]. This response is very interesting and clearly, additional studies, which are controlling/measuring the soil temperature in the rhizosphere, are required to determine the effect of irrigation and the irrigation procedure on the temperature of grape clusters.

### 3.2. Kaolin Film Transmittance and Reflectance Spectral Properties

To characterize the potential effects of kaolin applied to the leaves with regard to light intensity and quality, we performed transmittance and reflectance studies. For this purpose, the transmittance spectrum of a film of kaolin solution (5% w/v) on a glass plate was determined (Figure 4a). Although a high percentage of most photosynthetically active radiation (PAR) wavelengths is transmitted by the kaolin film, the blue light range is the least transmitted. The reflectance spectra obtained for leaves of grape plants sprayed with and without kaolin are represented in Figure 4b. Our results showed a relevant percentage of PAR is reflected by this white mineral, as compared to non-spayed (NK) vine leaves, rather than exclusively or mainly reflecting in the ultraviolet (UV) and infrared radiation (IR) ranges [47]. Additionally, kaolin was more efficient in reflecting UV light than IR light (in the measured ranges). These results are in accordance with previous studies using grapevine leaves [18] and other crops [27,28,48,49]. Thus, the beneficial effect of kaolin application is related to the reflection of excess radiation outwards, which reduces the risk of light stress-induced damage to leaves and fruit [47], while transmitting a very significant proportion of PAR.

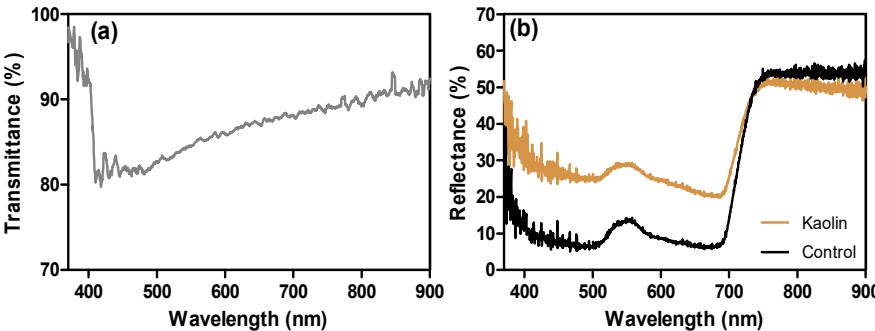

**Figure 4.** Transmittance (%) spectrum (**a**) of a kaolin suspension (5% w/v) and reflectance (%) spectra (**b**) of leaves with and without kaolin (control) (*n* = 3).

Together, these results call attention to the fact that foliar kaolin application may directly impact the photosynthesis of the sprayed leaves but also have an indirect effect on the non-sprayed leaves and grape berry clusters inside the canopy.

### 3.3. Effects on Berry Photosynthesis and Photosynthetic Pigments

#### 3.3.1. Maximum Quantum Efficiency of PSII

The maximum quantum efficiency of PSII ($F_v/F_m$) was determined *ex planta* under controlled conditions, using both exocarps and seeds from grape berries grown under the different treatments, microclimates, and three ripening stages (Figure 5). At the green stage, $F_v/F_m$ was similar in both berry tissues (Figure 5a,b). Upon further ripening, the exocarp kept its $F_v/F_m$ values (~0.7), while the seeds showed a significant decrease in this parameter, which reached values around 0.4–0.5 at the mature stage, which was in accordance with what was reported by Garrido et al. [24].

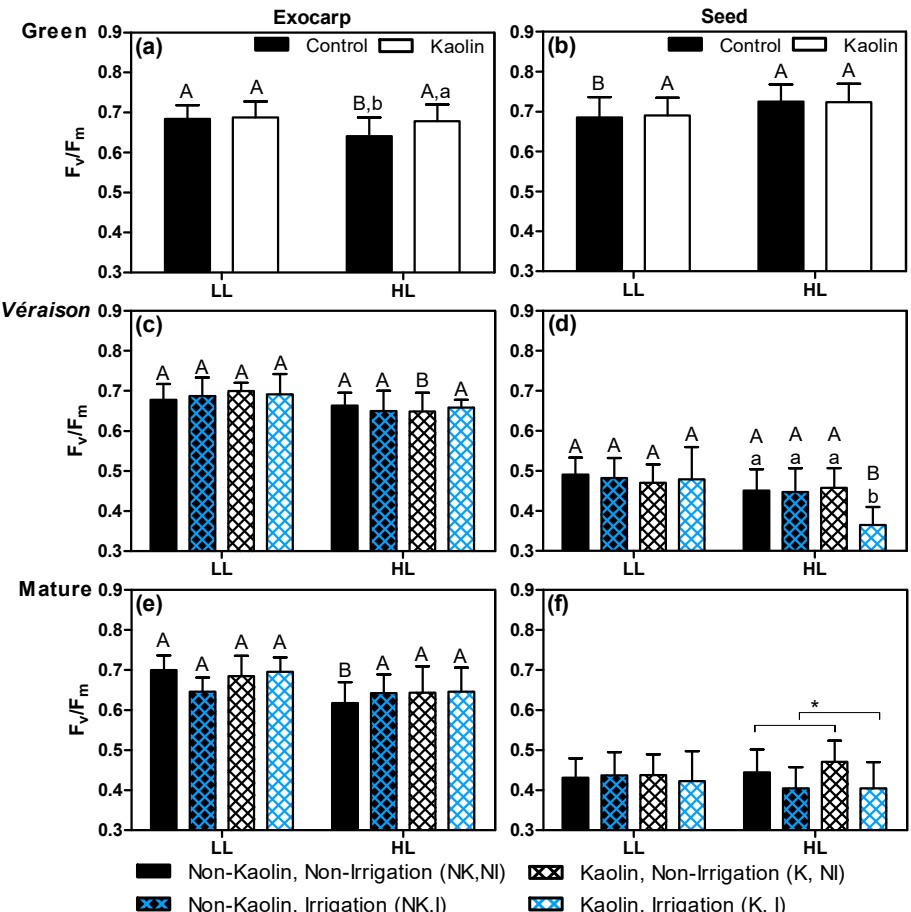

**Figure 5.** Maximum quantum efficiency of PSII ($F_v/F_m$) mean values (*n* = 12–24 berries, +SD) of exocarps and seed integuments obtained from dark-adapted LL and HL grape berries grown under the four combinations of the two treatments applied: irrigation (I)/ non-irrigation (NI) × kaolin (K)/ non-kaolin (NK). Samples were collected at three development stages (green, véraison, and mature). Statistical notation: for each developmental stage, capital letters refer to differences between light microclimates within the same treatment combination, and lowercase letters refer to differences between treatment combinations within each light microclimate (mean values with a common letter were not significantly different). When capital and lowercase letters are omitted, the respective factor did not have a significant effect (two-way ANOVA *p* > 0.05). Notation with an asterisk means that only one factor (kaolin or irrigation) was significant.

The cluster microclimate (LL vs. HL) had a significant effect on the $F_v/F_m$ values in both tissues and all developmental stages, with the exception of seeds at the mature stage, but exocarps and seed integuments responded differently to a light microclimate [24]. At the green stage, the exocarps from berries under control conditions, showed lower $F_v/F_m$ values in HL clusters, while their seeds showed significantly higher values than those from LL berries (Figure 5a,b). This was likely due to their inner location where the light transmitted through the skin and flesh tissues reaches values as low as 2% of the incident photon flux density (PFD) [50], which eventually translates a light limitation effect in LL clusters. These microclimate effects were more or less maintained in exocarps upon subsequent ripening (see NK + NI Figure 5a,c,e, while not being significant in véraison), while, in seeds, the difference between LL and HL clusters disappeared (Figure 5b,d,f), likely related to the (large) intrinsic ripening-dependent decrease in photosynthetic competence of this tissue.

Both kaolin and irrigation treatments of plants differentially influenced the $F_v/F_m$ values of the two berry tissues, with the seeds globally more responsive than exocarps, particularly to the irrigation treatment, which induced a significant effect at véraison and mature stages (Figure 5d,f). On the other hand, at these latter stages, no effects from treatments were detected on the $F_v/F_m$ of exocarps. This tissue only responded to kaolin treatment at the green stage, where HL berries showed an increment (6%) on $F_v/F_m$ by kaolin application (Figure 5a). In fact, the decrease in $F_v/F_m$ values of exocarps when comparing LL with HL berries at a green stage in control conditions (Figure 5a) had already been observed [24], which revealed that microclimates with higher luminosity can decrease $F_v/F_m$ values of exocarps at this stage. Together, these results suggest that foliar kaolin may protect the berry exocarp from excess light at a stage when the grape berry photosynthetic phenotype is still developing [24]. In HL seeds, the most prominent effect was a decrease in $F_v/F_m$ in irrigated-treated plants (Figure 5d,f). However, at véraison, this effect was observed only in kaolin-treated plants (Figure 5d). This apparent paradoxical effect may be related with the increased temperatures observed in irrigated grape berries in the hottest months (Figure 3c and Figure S1). In fact, the inhibitory effect of higher temperatures on $F_v/F_m$ was already reported for grapevine leaves [51].

### 3.3.2. Relative Rate of Electron Transport Through PSII (rETR$_{200}$)

The relative electron transport rate through photosystem II (rETR) was determined after acclimation of the berry tissues to an AL intensity of 200 μmol photons m$^{-2}$ s$^{-1}$ (rETR$_{200}$) in order to simulate average field light conditions.

At the green stage, the rETR$_{200}$ was higher in HL grape berries than in LL berries (both in control and kaolin-treated plants), especially in seeds (Figure 6a,b). Interestingly, a positive influence of kaolin application was observed in LL exocarps at this stage (Figure 6a), which suggests that more PAR reflected by kaolin is reaching the inside of the canopy, which may improve the exocarp photosynthesis of berries in shaded microclimates. At the véraison stage, this positive effect of kaolin was not detected, while it was evident at the mature stage (Figure 6e). In addition, a clear decrease in rETR$_{200}$ was observed in seeds, especially in HL clusters ($p < 0.0001$) during ripening (Figure 6b,d,f), which is similar to the results obtained for $F_v/F_m$. In addition, in seeds from HL-grown berries at the véraison stage of irrigated plants, the kaolin treatment led to a significant decrease in rETR$_{200}$ (Figure 6d: NK, I compared to K, I), which was observed for the parameter $F_v/F_m$ (Figure 5d). This pointed to a possible effect of berry temperature on seed photosynthesis. No reports were found for the effect of temperature on grape berry photosynthesis, but a continuous four-day exposure to high temperatures (38–40 °C) led to a decrease in photosynthetic activity of grapevine leaves [52–54].

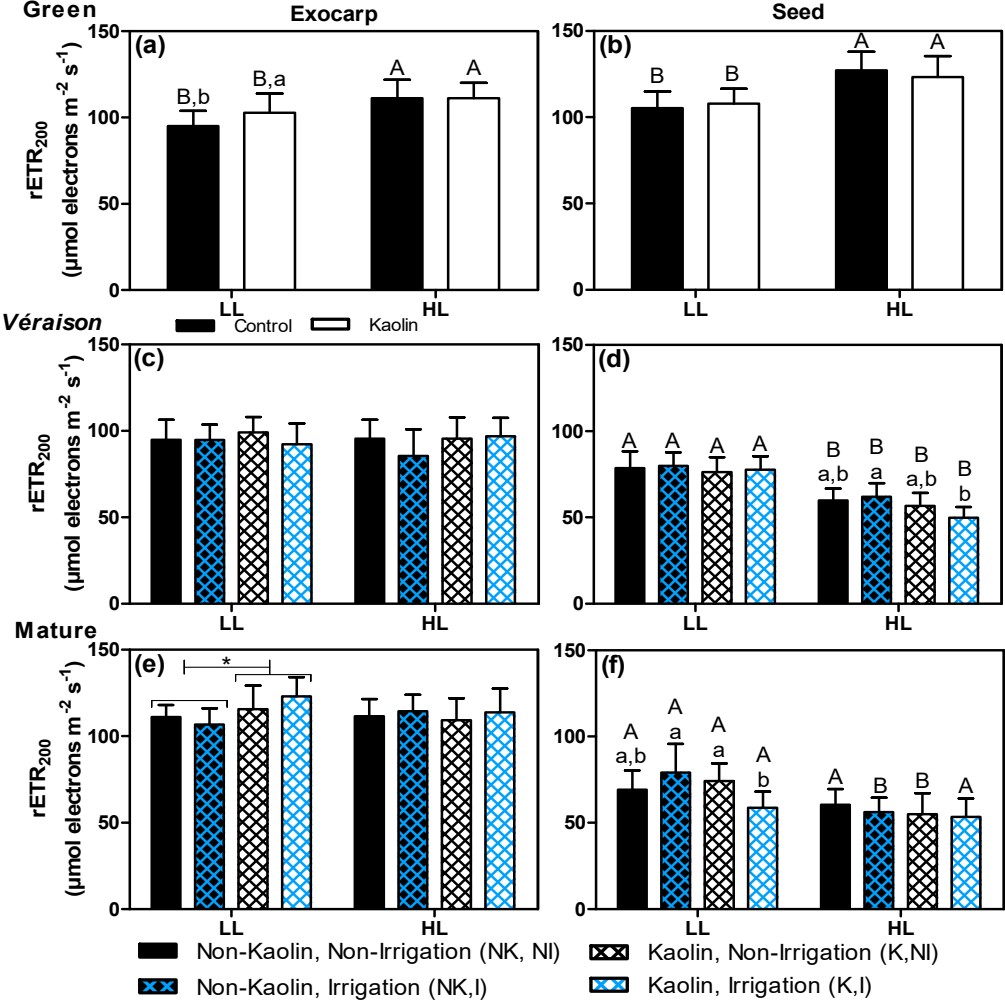

**Figure 6.** Mean values (*n* = 12–24 berries, +SD) of relative rate of electron transport through PSII at 200 μmol photons $m^{-2}$ $s^{-1}$ (rETR$_{200}$). All the microclimate conditions, treatment combinations and statistical information are the same as in Figure 5.

### 3.3.3. Non-Photochemical Quenching

A major component of non-photochemical quenching (NPQ) is the primary protective mechanism against light-induced photoinhibition, which involves various processes dissipating excessive non-radiative energy [55], including the xanthophylls cycle (e.g., as shown in grapevines, [56]) and phosphorylation/dephosphorylation of light harvesting complexes [57].

The NPQ results are represented in Figure 7. When comparing the berry tissues, it can be concluded that the exocarp tissue consistently exhibits roughly two-fold higher NPQ values than seeds. This result suggests that the exocarp exhibits more developed mechanisms of photoprotection, which is consistent with the fact that it is an external, more exposed tissue. During berry ripening, NPQ values decreased in both tissues, especially in seeds. Seeds attained very low NPQ values at later stages, which is in line with their $F_v/F_m$ (Figure 5b,d,f) and rETR$_{200}$ (Figure 6b,d,f) profiles, which likely reflect the normal ripening-related loss of photosynthetic functioning of seeds. For the exocarps, dissipation or quenching mechanisms other than NPQ may explain the result, since this tissue maintains high photosynthetic activity until the mature stage (Figure 6a,c,e). Accumulation of carotenoids in white berries ('Sauvignon Blanc') was also increased in response to increasing levels of solar light in the canopy, which shows that the berries utilize these photosynthesis-related pigments in photo-acclimation responses and/or as "sunscreens" [58].

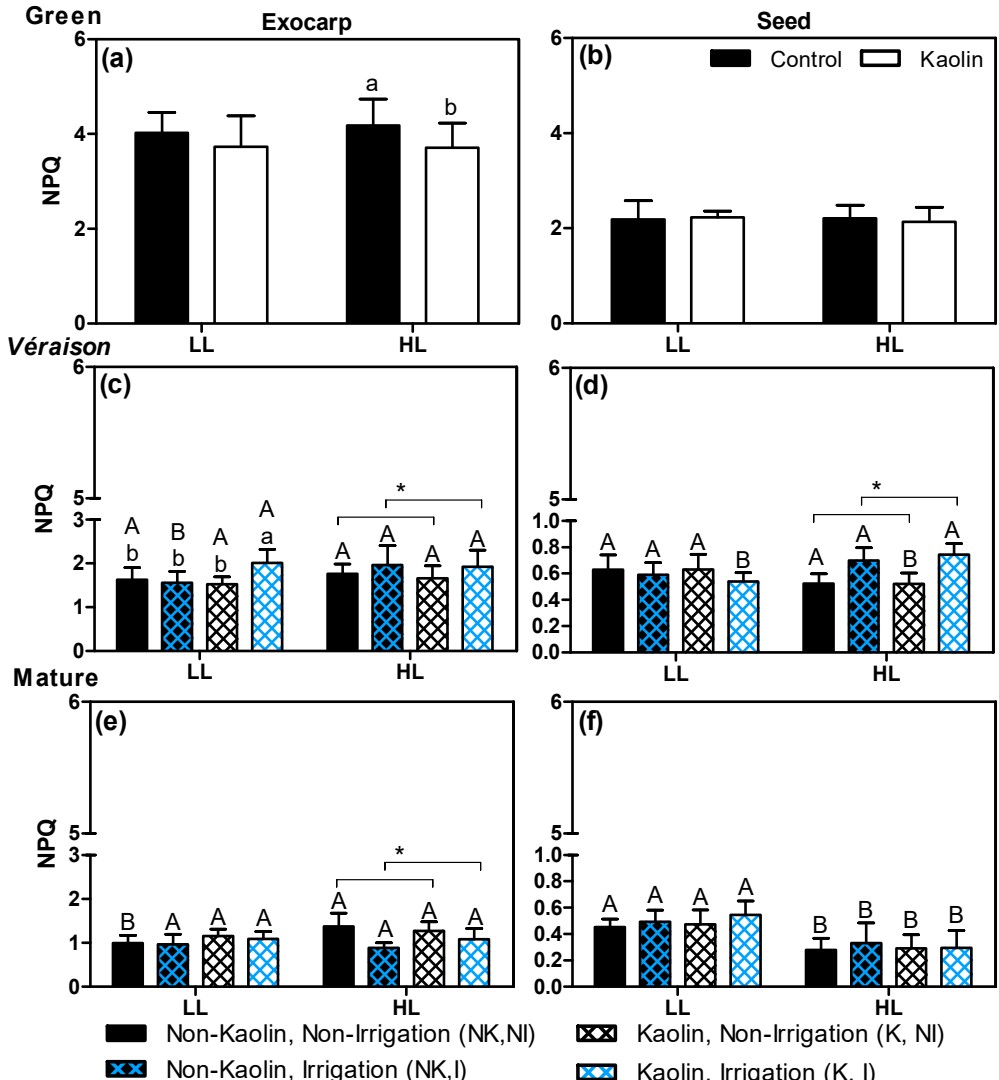

**Figure 7.** Non-photochemical quenching (NPQ) mean values (*n* = 12–24 berries, +SD). All the microclimate conditions, treatment combinations, and statistical information are the same as in Figure 5.

Regarding the effect of treatments, we found that the foliar kaolin application promoted lower NPQ in HL exocarps at the green stage, when compared with their NK controls (Figure 7a), which suggests that kaolin helps to protect these HL berries from excessive radiation absorption. This is similar to the $F_v/F_m$ results (Figure 5a).

At the véraison stage, both HL exocarps and HL seeds showed increased NPQ in irrigated-treated grape berries (Figure 7c,d). In LL exocarps, NPQ was also increased in K,I berries, when compared to the remaining treatment combinations (Figure 7c). This increase in NPQ values of grape berries in irrigated plants, was lost at the mature stage, even though it is important to note that NPQ values were already very low at this stage (Figure 7e,f). This irrigation-related feature had already been observed with other parameters and is discussed above. The increased temperatures registered at later developmental stages of grape berries (Figure 3c), can impose other limitations or impairments, and, thus, recruit more energy-dissipation by NPQ, and eventually by other dissipative mechanisms.

### 3.3.4. Photosynthetic Pigments

To better evaluate the impact of foliar kaolin application on the light microclimate of grape berry clusters and its relationship with berry photosynthesis, and non-photochemical mechanisms, photosynthetic pigments were quantified in exocarps and seeds of both LL and HL-exposed grapes.



Results obtained for the green stage are depicted in Figure 8 (for later stages, see Supplementary Materials). At control conditions, the HL berries had higher levels of both chlorophylls and carotenoids than LL berries, in both tissues. Additionally, and in line with the rETR$_{200}$ results (Figure 6a), kaolin application resulted in a marked increase by 26% in chlorophylls and 82% in carotenoids content in exocarps from LL berries (Figure 8a,c), which support the idea that more light reached the inner parts of the kaolin-sprayed canopy. This is fundamental to build the photosynthetic machinery [59]. During ripening, the photosynthetic pigments decrease in both tissues and especially in the seed integuments (Figures S3 and S4) and no consistent and conspicuous effects by combined mitigation treatments were observed (Figure S3).

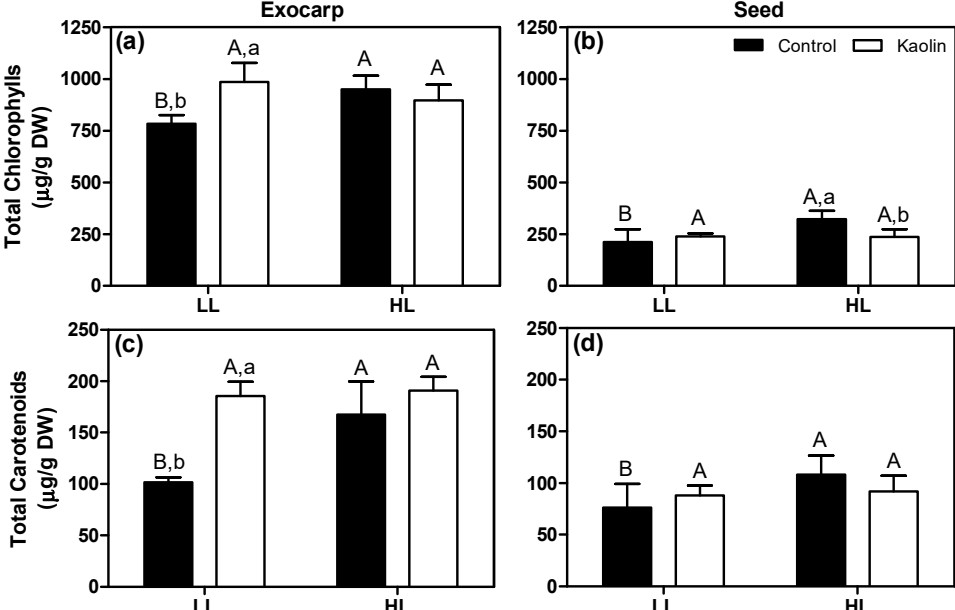

**Figure 8.** Chlorophylls (**a,b**) and carotenoids (**c,d**) concentration mean values (*n* = 3, +SD) of exocarps and seeds obtained from LL and HL grape berries grown under non-kaolin (black columns) and kaolin (white columns) application, and collected at the green stage. Statistical notation: different capital letters refer to significant differences (two-way ANOVA, *p* ≤ 0.05) between the two light microclimates within the same plant treatment, and different lowercase letters to differences between treatments within each light microclimate. If the respective factor did not have a significant effect, the letters were omitted.

In addition, and supporting the view discussed above, the higher grape berry temperature was registered in irrigated treatments at later developmental stages (Figure 3c), by imposing physiological impairments. The temperature recruits more energy-dissipation by NPQ (Figure 7), which is the fact that carotenoids contents (Figure S4), but not chlorophylls (Figure S3), were also increased by irrigation treatment, for both tissues at the véraison stage.

Overall, the results obtained by pulse amplitude modulated fluorometry showed that, for the external tissue, exocarp, and foliar kaolin application led to an increase of $F_v/F_m$ (Figure 5a, HL), rETR$_{200}$ (Figure 6a,e LL), and a reduction in non-photochemical quenching (Figure 7a, HL). To our best knowledge, this is the first work assessing the impact of foliar kaolin application on photochemical and non-photochemical functions in grape berries. Recently, it was verified that grapevine leaves with kaolin display the same response, i.e., an increase in $F_v/F_m$, $\Phi_{II}$, and ETR, and a decrease in NPQ [18,60]. Similar results were also reported for olive leaves [61]. In this way, and in terms of photochemical processes, those kaolin-treated leaves have lower photo-inhibitory damage [17,62], and the open PSII reaction centers captured the light absorbed by PSII antenna more efficiently [17,35]. This response

was likely due to a reduced loss of excitation energy by thermal dissipation, which could compete with its transference to PSII reaction centers, as shown by the lower NPQ values [17,35].

For exocarps of grape berries growing in inner parts of the canopy (LL microclimate), the photosynthetic results revealed that foliar kaolin application, may cause an extra "sunscreen" effect, and did not have a negative effect on those parameters, which we conjectured in our previous work [24]. The increased reflection provided by this mineral to inner parts of the canopy allowed good photochemical performance of LL exocarps, which is contrary to what we hypothesized in our previous work [24]. This contributes to higher carbon gains at the whole canopy level and also at the fruit level.

Regarding the results for the seed integument (internal organ), the positive effects of kaolin were observed mainly in non-irrigated plants such as an increase in $F_v/F_m$ (Figure 5f, HL) and a decrease in NPQ (Figure 7d, HL). In more temperate or Mediterranean regions, this seems like a positive effect, but these results also show the importance of the irrigation system. The interaction between kaolin application and irrigation treatments on grapevine leaves have been studied before [42,63–66]. However, based on our knowledge, no study has approached the impacts on photosynthetic activity at the grape berry level, using chlorophyll fluorescence analysis.

## 4. Conclusions

The purpose of the current study was to assess the effects of foliar application of kaolin and irrigation, as abiotic stress mitigation strategies, on the photosynthetic activity of exocarps (skins) and seeds of grape berries growing under different light microclimates in the canopy. One of the most relevant findings was that the kaolin applied to leaves increased the photosynthetic activity of both exocarps and seed integuments of berries growing under low light conditions in the canopy. This is likely due to higher reflection of PAR to the inner zones. We believe, though, that the beneficial effects will depend on the canopy structure and on the incident radiation, with denser canopies and higher radiations conferring higher overall photosynthetic gains. Somewhat puzzling was the observation that seeds of irrigated plants showed lower photosynthetic activities, in the véraison and mature stages, especially under kaolin treatment. Several causes may explain this unexpected phenomenon, so more detailed and ad-hoc design studies should be conducted to address this relevant finding.

This comprehensive study provides the first evidence of foliar kaolin application as a procedure allowing the modulation of photosynthesis in the grape berry, but also calls attention to the importance of the irrigation system. In this way, this knowledge can be used by farmers to support their decisions concerning sustainable adaptation strategies applied on vineyards. Research to unveil the function of berry tissues' photosynthesis on the metabolome of the grapes is already underway, which ultimately contributes to the final quality of the fruit and wine.

**Supplementary Materials:** The following are available online at http://www.mdpi.com/2073-4395/9/11/685/s1. Figure S1. Meteorological elements from IPMA Institute from Braga city. (a) Temperature (°C) maximal, average, and minimal. (b) Total precipitation (mm). Figure S2. Temperatures of full exposed leaves at the green stage (a,b) and the mature stage (c,d), for plants with irrigation (blue columns, note: the textured blue columns at green stage i.e., before the onset of irrigation, represent the measurements in the plots that were later irrigated) and foliar kaolin application (white columns). Black columns correspond to the respective controls. Values represent means with standard deviation (*n* = 16 plants). Statistical notation: per ripening stage, different lowercase letters refer to significant differences (*p* ≤ 0.05) between treatments. Whenever letters are omitted, it means that the respective factor did not have a significant effect. Figure S3. Chlorophylls concentration mean values (*n* = 3, +SD) of exocarp and seed obtained from LL and HL grape berries grown under the four combinations of the two treatments applied: irrigation (I)/ non-irrigation (NI) × kaolin (K)/ non-kaolin (NK). Samples were collected at three development stages (green, véraison, and mature). Statistical notation: for each developmental stage, capital letters refer to differences between light microclimates within the same treatment combination, and lowercase letters refer to differences between treatment combinations within each light microclimate (mean values with a common letter were not significantly different). When capital and lowercase letters are omitted, the respective factor did not have a significant effect (two-way ANOVA *p* > 0.05). Figure S4. Carotenoids concentration mean values (*n* = 3, +SD) of exocarp and seeds. All the microclimate conditions, treatment combinations, and statistical information are the same as in Figure S3.

**Author Contributions:** Conceptualization, A.G., A.C. (Artur Conde), R.V., and A.C. (Ana Cunha) Methodology, A.G., J.S., and A.C. (Ana Cunha) Formal analysis, A.G. and A.C. (Ana Cunha). Investigation, A.G. and A.C. (Ana Cunha). Resources, J.S. and R.V. Writing—original draft preparation, A.G. Writing—review and editing, A.C. (Ana Cunha), J.S., R.V., and A.C. (Artur Conde). Supervision, A.C. (Artur Conde), R.V., and A.C. (Ana Cunha) Project administration, A.C. (Ana Cunha).

**Funding:** The FCT-Portuguese Foundation for Science and Technology by the grant provided to Andreia Garrido (PD/BD/128275/2017), under the Doctoral Program "Agricultural Production Chains – from fork to farm" (PD/00122/2012), funded this research and APC.

**Acknowledgments:** The National Funds by FCT - Portuguese Foundation for Science and Technology, under the strategic programmes UID/AGR/04033/2019 and UID/BIA/04050/2019, and the project "INTERACT - VitalityWine - NORTE-01-0145-FEDER-000017 – funded by Norte2020 supported the work. The FCT and FEDER/COMPETE/POCI - Operational Competitiveness and Internationalization Program, under Project the projects MitiVineDrought – PTDC/BIA-FBT/30341/2017 (POCI-01-0145-FEDER-030341), and POCI-01-0145-FEDER-006958 also supported this work. Artur Conde was supported with a post-doctoral fellow of the mentioned INTERACT/VitalityWine project with the Reference BPD/UTAD/INTERACT/VW/218/2016, and also supported by a post-doctoral researcher contract/position within the project "MitiVineDrought" (PTDC/BIA-FBT/30341/2017 and POCI-01-0145-FEDER-030341). This work also benefited from the networking activities within the European Union-funded COST Action CA17111 – "INTEGRAPE - Data Integration to maximize the power of omics for grapevine improvement". Authors acknowledge the owner from *Quinta Cova da Raposa*, Manuel Taxa, who provided the samples, Susana Chaves (from CBMA) for her English grammar revision, and also all support given by the Biology Department of the School of Sciences from the University of Minho.

**Conflicts of Interest:** The authors declare no conflict of interest.

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
