# Peer review of "Influence of Foliar Kaolin Application and Irrigation on Photosynthetic Activity of Grape Berries"

_agronomy, doi:10.3390/agronomy9110685_

Round 1

Reviewer 1 Report

The authors followed all the suggestions raised and the manuscript in the present form resulted improved.

I am still not convinced on the reply to my previous comment regarding the details reported for the irrigation treatment. Even if I understand that the study was not mainly addressed on drought, "irrigation" is mentioned even in the ms Title and even if the authors added more details, I believe that the critical information is still lacking. In particular they reported "Water was applied by drip irrigation with one dripper per vine and a drip line (2 cm diameter) placed approximately 80 cm above the soil. Irrigation occurred every 3 days, once a day either early in the morning or late in the afternoon, for 2 hours with an average water flow of 70 L h-1 vine-1". But, if I am right, this means that each plant would have received a water amount of about 140 L!!! That sounds me extremely high!! The only critical parameter that is still lacking is the dripper capacity, authors must provide this information instead of the drip line diameter.

Author Response

The authors thank the careful revision and the improvement introduced on the manuscript.

The reviewer is completely right about his/hers concern related to the amount of water received by each vine per hour. We apologize for this mistake and indeed appreciate the correction. This water flow of 70 L h-1 vine-1 was determined in the main pipe, that is, the water flow that is distributed to each row, and then divided by the number of drippers/vines.

Now, and according to the reviewer suggestion, we determined the mean value of water debit by dripper (mean dripper capacity, in L h-1, considering measurements in 12 randomly selected drippers), in order to have an accurate value of the amount water received by each vine. In this way, we changed this information in the topic 2.1 of the Material and Methods section to: “Water was applied by drip irrigation with one dripper per vine and a drip line placed approximately 80 cm above the soil. Irrigation occurred every 3 days, once a day either early in the morning or late in the afternoon, for 2 hours, with an average dripper capacity of 5.5 ± 1.6 L h-1 (n = 12 randomly selected drippers, ± SD).” (lines 116 to 119).

Reviewer 2 Report

No comments

Author Response

The authors thank the revision and the global appreciation of the manuscript.

This manuscript is a resubmission of an earlier submission. The following is a list of the peer review reports and author responses from that submission.

Round 1

Reviewer 1 Report

The paper is clear and concise, and provides valuable insight for mitigation strategies which can be implemented in grapevine cultivation in terms of climate changing. However, the description needs minor improvement.

From the text I am not completely sure about non-kaolin application. Were the leaves sprayed with water or left untreated? How the kaolin was sprayed, by hand? What about the berries, were they sprayed occasionally during the procedure?

Why light intensity and temperature was measured between 15h and 17h?

2.4. Dark cabinet or dark chamber would be more appropriate than "dark room".

2.6. "Analysis of variance" is proper.

Figures: while using post-hoc test, A (a) letter should be designed for the highest value(s).

Supplementary material, Fig. S2 - there is no capital letter in the figure despite the description in the caption.

Reviewer 2 Report

The present ms aims at studying the effect of foliar kaolin treatment and irrigation practice on grape berry photosynthesis under low light and high light canopy microclimates. I have main concerns regarding the study proposed mainly driven by the bad overall presentation and english editing and on methodological lacks. In particular more accurate and precise use of phenological stages (i.e. BBCH) for both the vegetative and reproductive growth stages must be used in order to allow the full understanding and repeatability of the trials. Furthermore no information at all are reported for the irrigation treatment, except for the pipe system (I hope that a drip irrigation was used) of 140 l h-1! The amount and duration of the irrigation was based on what? Does any physiological measurements has been done in order to check the occurred or not drought and its level? If none of these data are available it is difficult to understand vine's responses except from a very empirical and superficial basis.

Most of the histogram charts need a clear legend to be inserted since it was really challenging for me to understand the different treatment and the specific control ones. In some cases, based on the size of the standard deviation above and below the mean values, it is difficult to me to understand the significant level of the differences reported by the letters. And regarding this, what does means that "posthoc multiple comparisons were performed using the Bonferroni test?" If I am right, a specific Bonferroni correction must be applied if multiple and subsequent comparison are applied and this in my opinion is not the case, causing a lower p-values (i.e. than 0.05) for considering as significant the differences among the treatments.

Minor comments:

l52: please consider more updated IPCC reports;

l65: I consider this sentence on stomata closure and photosynthesis decrease, too general and it cannot be reported within the more specific climate change impacts listed.

l84: canopy structure strategies?

l102: recently projection model...

l107-111: I do not believe irrigation could improve leaf photosynthesis, but only can be used to avoid water stress.

l138: please pay attention since here and throughout the entire ms many figures are missing the citation ref due to such errors.

l140: please use a phenological stage code (i.e. BBCH)

l141-142: see the comments above for the irrigation treatment;

l140: the irrigation was applied on half of the block plants? How? So on two of four plants per block? Is the block made of four continuous vines in the row?

l146: subclusters? This mean a small cluster part was sampled? I believe that the position of the sample within the cluster is crucial in this case and more than a randomly chosen procedure at least two cluster faces have been sampled and treated separately.

l149: please use a phenological stage code (i.e. BBCH)

l162: please use the propoer unit for light intensity, are you talking of µmol of photons m-2 s-1? PPFD values of 200 µmol of photons m-2 s-1 for exposed berries to direct radiation in summer days sounds me too low.

l170: please add a reference to this specific protocol used as done in the other paragraphs (2.2, 2.4, 2.5).

l239: Climateric?

l239: wine-growin? Please correct here and use growing season instead throughout the ms.

l255-259: this sentence in unclear

Figure 2: please see the comments reported above.

l260: I am not convinced of the significance of these reported small differences

l280: bad english

l305: don't you believe that this sentence is a bit too speculative?

l314: even if you reported that irrigation was applied early in the morning or late in the afternoon?
